# Comparisons of Dental Anomalies in Orthodontic Patients with Impacted Maxillary and Mandibular Canines

**DOI:** 10.3390/diagnostics13172766

**Published:** 2023-08-25

**Authors:** Joanna Stabryła, Małgorzata Zadurska, Paweł Plakwicz, Krzysztof Tadeusz Kukuła, Ewa Monika Czochrowska

**Affiliations:** 1University Medical Center, Medical University in Warsaw, 02-097 Warsaw, Poland; j.abramczykk@wp.pl; 2Department of Orthodontics, Medical University in Warsaw, 02-097 Warsaw, Poland; malgorzata.zadurska@wum.edu.pl; 3Department of Periodontology, Medical University in Warsaw, 02-097 Warsaw, Poland; info@plakwicz.com; 4Department of Oral Surgery, Medical University in Warsaw, 02-097 Warsaw, Poland; kkukula@gazeta.pl

**Keywords:** canine transmigration, dental anomaly, impacted mandibular canine, impacted maxillary canine, tooth agenesis, tooth impaction

## Abstract

To assess the presence and possible associations between the type of dental anomalies and maxillary and mandibular canine impactions in orthodontic patients treated for canine impaction, panoramic radiographs of orthodontic patients treated for canine impaction were assessed for the presence of associated dental anomalies. A random sample of orthodontic patients without canine impaction matched for age and gender served as controls. Descriptive and exact inferential statistics were implemented in order to assess potential associations between canine impaction and dental anomalies. A total of 102 orthodontic patients with 70 maxillary (MaxCI) and 32 mandibular (ManCI) canine impactions were assessed. The control group included 117 orthodontic patients. Dental anomalies were present in more than 50% of patients with impacted canines and in 20% of the controls. Tooth agenesis was significantly more common in the MaxCI group when compared to the ManCI group, while supernumerary teeth and canine transmigration were registered more often in the ManCI group. When compared to the control group, peg-shaped maxillary lateral incisors and tooth agenesis were significantly more prevalent in the MaxCI group, while canine transmigration, supernumerary teeth, the agenesis of mandibular incisors and tooth transpositions were significantly more prevalent in the ManCI group. The impaction of other teeth was significantly more common in both canine impaction groups when compared to the controls. The prevalence of dental anomalies in orthodontic patients with impacted canines was higher than in orthodontic patients without canine impaction. Different types of tooth anomalies were found in the MaxCI and ManCI groups.

## 1. Introduction

The prevalence of maxillary canine impaction in Caucasians has been reported to range from 1 to 3.5%, and it is the second most common tooth impaction after third molar impaction [1,2,3,4]. The discrepancy in prevalence between studies may be attributed to population and methodological differences. Most of impacted maxillary canines are positioned palatally and canine impaction is twice as common in women when compared to men [1,4]. Impacted canines may pose a functional and an esthetic problem for the patient and can affect neighboring incisors [5]. Orthodontic treatment for severely impacted maxillary canines is usually long and complicated, therefore early detection of canine impaction is important for the successful implementation of interceptive methods, which may facilitate a spontaneous correction of the ectopic position and a normal canine eruption. It has been suggested that maxillary arch expansion or extraction of a maxillary primary canine also combined with a first primary molar extraction may help the normal eruption of the impacted maxillary canine [6,7,8]; however, the effectiveness of this approach has been contested in more recent reports [9].

Canines follow a long eruption path and a correct diagnosis of future canine impaction in young children may be clinically and radiologically challenging. Orthodontists often detect the ectopic position of a canine bud which may lead to its future impaction. The possible role of accompanying dental anomalies in the etiology of maxillary canine impaction has been reported in the literature since the late 1960s [10]. The early detection of potentially related dental anomalies to an ectopic canine position can be an additional diagnostic and predictive tool for canine impaction. Table 1 summarizes the existing literature on the association between dental anomalies and maxillary canine impaction. The most commonly reported dental anomalies associated with maxillary canine impactions are peg-shaped and agenesis of maxillary lateral incisors, the agenesis of second premolars and other teeth, tooth transposition and other tooth impactions.

The prevalence of mandibular canine impaction is much lower than that of the upper canines [25] and it is estimated to range between 0.3 and 1.35% (Table 2). 

Buccal location is more common than a lingual location. Preventive treatment is usually not possible for the severely impacted mandibular canine, since there is no evidence that primary mandibular canine extraction leads to a spontaneous eruption of the permanent successor. The orthodontic expansion or extrusion of severely impacted and tilted mandibular canines is difficult and often impossible because of the morphology of the alveolar process and the position of neighboring teeth [26,30]. The extraction of the severely impacted lower canine has been reported to be the most common treatment alternative [29,34,38].

Up until now, only a few studies have reported on dental anomalies related to mandibular canine impaction, and they are mostly related to the transmigration of an impacted canine (Table 2). No comparisons regarding the type of dental anomalies between impacted maxillary and mandibular canines have yet been reported, which may possibly give an insight into the etiology of canine impaction and the implementation of interceptive treatment or surgical uprighting.

Therefore, the aim of the present study was to assess the presence and possible associations between the type of dental anomalies and maxillary and mandibular canine impactions in an attempt to clarify whether these can be used as a diagnostic tool for canine impaction during a radiological screening in young patients.

## 2. Materials and Methods

Panoramic radiographs in which canine impaction or a severe ectopic position were diagnosed were selected from the files of three dental offices (EC, KK, PP). The inclusion criteria included all available orthodontic patients treated for maxillary or mandibular canine impaction. The exclusion criteria included patients with dentofacial deformities and patients with canine impaction in both jaws, as the presence of a specific dental anomaly could not be directly related to the location of the canine impaction. The diagnosis of canine impaction was suspected based on the absence of a permanent canine in the oral cavity at least 6 months after the eruption of the corresponding tooth on the other side and after its root development was completed, or if a severe distal angulation or ectopic position of an unerupted canine was present on radiographs [39]. The additional radiological examination included a cone-beam computed tomography to precisely locate the ectopic canine using a VeraviewPoc (Morita Corp., Kyoto, Japan) with a voxel size of 0.125 mm and a slide thickness/interval of 0.125 mm. All patients included in the study were treated for the canine impaction using interceptive treatment, orthodontic extrusion, transalveolar autotransplanation or canine extraction [40].

The panoramic radiographs were assessed for the presence of dental anomalies by one independent rater, a specialist in orthodontics, who was not involved in the treatment of the patients (JS). The rater was calibrated to assess the presence of dental anomalies with a senior orthodontist (EC). The presence of dental anomalies was confirmed with two other authors (PP, EC), when necessary. Patients’ age, gender and the localization of the impacted canines (buccal, central, palatal/lingual) were also recorded. Panoramic radiographs of orthodontic patients without canine impaction from the office of one of the authors (EC) matched for age and gender served as the control. The controls were recruited from a dental practice limited to orthodontics which had been run for over 25 years.

The patients were assessed for the presence of the following dental anomalies: peg-shaped maxillary lateral incisors and their agenesis, the agenesis of second premolars, the agenesis of mandibular incisors, any tooth agenesis, supernumerary teeth, tooth transposition, canine transmigration and impaction of other teeth. The presence of a specific dental anomaly was confirmed using other available orthodontic records (study models, photographs, all available radiographs), in the presence of uncertainty. The diagnosis of peg-shaped incisor was performed according to Becker et al. [41]. The agenesis and impaction of wisdom teeth was not included in the analysis because this cohort included patients younger than 11 years old and the detection of third molar formation is uncertain at that age [23]. The specific dental anomaly was registered, if present unilaterally or bilaterally per patient.

### Statistical Analysis

Descriptive statistics and frequency distributions of anomalies were calculated for the impaction and control groups and by jaw within each impaction group. Differences in baseline characteristics between patient groups were assessed with an independent t-test, Χ^2^ or Fisher’s exact test depending on the outcome and the event frequency. A univariable exact logistic regression was implemented to assess potential associations between type of anomalies, group and jaw. Odds ratios, associated 95% confidence intervals and *p*-values were calculated. Statistical significance was set at 0.05, and all analyses were conducted using Stata 14.1 statistical software (Stata Corp, College Station, TX, USA).

## 3. Results

Four patients with impacted canines in both jaws were excluded from the study. Therefore, 102 patients with 70 maxillary and 32 mandibular canine impactions were assessed. The patient flow diagram is presented in Figure 1. A total of 51 females (72.86%) and 19 males (27.14%) presented with impacted maxillary canines, and 19 females (59.38%) and 13 males (40.63%) with impacted mandibular canines. The control group included 117 orthodontic patients without canine impaction, 81 females (69.23%) and 36 males (30.77%).

The mean age in the maxillary canine impaction group (MaxCI) was 20.6 years old (SD: 10.32; range: 10.13–60.02; median: 15.7; interquartile p25: 13.4; interquartile p75: 27.4), and in the mandibular canine impaction group (ManCI), it was 15.2 years old (SD: 9.3, range: 7.5–53.9; median: 12.3; interquartile p25: 10.2; interquartile p75: 15.2). The mean age in the control group was 17 years old (SD: 10.81; range: 7.7–61.1; median: 12.7; interquartile p25: 10.3; interquartile p75: 19.2).

Twelve patients had a bilateral impaction of maxillary canines, and four patients had a bilateral impaction of mandibular canines. In the MaxCI group, 51 canines (72.9%) were palatally impacted, 10 canines (14.3%) were bucally impacted, and 9 canines (12.9%) were located in the middle of the alveolar process. In the ManCI group, 25 canines (78.1%) were buccally impacted, 2 canines (6.25%) were lingually impacted, and 5 canines (15.6%) were located in the middle of the alveolar process.

### 3.1. Prevalence of Dental Anomalies in Impaction and Control Groups

The prevalence of dental anomalies in relation to the number of their occurrence in the impaction and control groups is shown in Table 3. Dental anomalies were registered in 52.9% of the patients in the MaxCI group and in 53.1% in ManCI group. In the control group, dental anomalies were present in 20.1% of the patients. The difference was statistically significant for both maxillary and mandibular canine impactions versus controls (MaxCI: *p* < 0.001, ManCI: *p* < 0.01); however, the difference was not significant between the MaxCI and the ManCI groups (*p* = 0.98).

### 3.2. Comparisons of Dental Anomalies between Maxillary and Mandibular Canine Impaction Groups

The prevalence of the various dental anomalies in the MaxCI and the ManCI groups is presented in Table 4. The prevalence of dental anomalies in the MaxCI group ranged from 0 for the presence of canine transmigration to 41.4 percent for the presence of any tooth agenesis. In the ManCI group, the prevalence ranged from 3.1 percent for the presence of a peg-shaped maxillary lateral incisor and an agenesis of the second premolar and mandibular incisor to 18.8 percent for the presence of any tooth agenesis, supernumerary tooth and canine transmigration. The prevalence of any tooth agenesis was significantly higher in the MaxCI group when compared to the ManCI group, whereas the prevalence of a supernumerary tooth and canine transmigration was significantly higher in the ManCI group when compared to the MaxCI group.

### 3.3. Comparisons of Dental Anomalies between the Canine Impaction and the Control Groups

The prevalence of the various dental anomalies in the MaxCI, the ManCI and control groups is shown in Figure 2 and in Table 5. The prevalence of peg-shaped maxillary lateral incisors and any tooth agenesis was significantly higher in the MaxCI group, whereas patients with ManCI had a significantly higher prevalence of agenesis of a lower incisor and the presence of a supernumerary tooth, tooth transpositions and canine transmigration when compared to the control group. Patients in both groups with canine impaction had a higher prevalence of impaction of other teeth than controls.

## 4. Discussion

This study assessed the presence and the type of dental anomalies in orthodontic patients with maxillary and mandibular canine impactions. The aim was to gain insight into the early detection of canine impaction based on the presence of other dental anomalies. All patients included in the study group were treated for maxillary or mandibular canine impactions and therefore, impaction was unequivocally confirmed. Three patients with mandibular canine impaction, who were younger than 10 years old when the first panoramic radiograph was taken, had a severe ectopic position of a mandibular canine and were later treated by transalveolar transplantation of the ectopic canines since orthodontic extrusion was not possible. All other patients were older than 10 years old. The first available panoramic radiographs were chosen for the analysis of other associated dental anomalies as they were routinely obtained at the initial orthodontic patient screening.

The control group consisted of the orthodontic patients having no canine impaction. The inclusion of patients seeking orthodontic treatment as control is likely to make the cohorts more homogeneous in other characteristics, since canine impaction is more common in this group than in the normal population [35]. We excluded patients with canine impaction in both jaws because of the small sample size (four patients) and the difficulty to relate the presence of the anomaly to the location of an impacted canine. It would be interesting to compare groups with maxillary and mandibular canine impactions and canine impactions in both jaws regarding the presence of dental anomalies. However, the concomitant impaction of maxillary and mandibular canines is rare, and it is difficult to collect an adequate sample.

In the present study we included a relatively high number of impacted mandibular canines. Celikoglu et al. [31] found nine impacted mandibular canines among 2215 orthodontic patients, and four impacted mandibular canines in 453 orthodontic patients were reported by Kamiloglu and Kelahmet [37]. Only one more study from Table 2 included a cohort of 20,347 orthodontic and pediatric patients, which was also the largest cohort included, and 64 mandibular canines were identified [38]. As shown in Table 2, the existing evidence on the prevalence of dental anomalies in patients with impacted mandibular canines is very limited and so far, no comparisons between the prevalence of dental anomalies in patients with impacted maxillary or mandibular canines have been performed.

Dental anomalies were significantly more frequent (>50%) in both groups of patients with impacted canines in comparison to patients without canine impaction (20%). This is an important finding regarding the early detection of canine impaction in young patients, possibly suggesting the need for interceptive treatment. No significant difference in the prevalence of dental anomalies between patients with impacted maxillary and mandibular canines was found; however, different types of dental anomalies were detected between those groups.

In our material, maxillary canine impaction was found to be more common in orthodontic females than in males, and this is in agreement with previous studies (Table 1). Therefore, the implementation of preventive methods in orthodontic female patients suspected of maxillary canine impaction and in the presence of other dental anomalies perhaps should be considered. It is interesting that no such distinct gender difference was found for mandibular impaction, and this is also in agreement with some of the previous studies (Table 2).

### 4.1. Comparisons of the Presence of Different Dental Anomalies between Groups

Ectopic maxillary first molar eruption is associated with an increased risk of ectopic canine eruption [12,42]. We did not observe any signs of ectopic eruption of maxillary first molars in any of the examined patients; therefore, this dental anomaly was not assessed. This may be related to the inclusion of older patients in our sample in whom such eruption disturbance was not possible to detect.

### 4.2. Peg-Shaped Maxillary Incisors and Agenesis of Lateral Incisors

The prevalence of peg-shaped maxillary lateral incisors in our study was significantly higher in the MaxCI group in comparison to the control group, while the agenesis of maxillary lateral incisor did not differ between those groups. The canine guidance theory related disturbances in maxillary canine eruption to the absence of a neighboring lateral incisor root [43]. However, Peck et al. [14], Chaushu et al. [16] and Yan et al. [21] reported no difference in the prevalence of an agenesis of maxillary lateral incisors in patients with palatally displaced canines. At the same time, they reported significantly smaller dimensions of maxillary lateral incisors, which were also present in our material. The shape and the number of maxillary lateral incisors did not significantly differ between patients with maxillary versus mandibular canine impaction. This is an interesting finding because the agenesis of maxillary lateral incisors cannot be not associated with the eruption path of the mandibular canine.

### 4.3. Tooth Agenesis

Tooth agenesis was documented to be strongly associated with maxillary canine impaction in a number of studies [44] (Table 1). The results of the present study support the findings for the occurrence of tooth agenesis in the MaxCI group, but not specifically for the presence of maxillary lateral incisor agenesis or second premolar agenesis. All adult patient (three patients) who were diagnosed with premolar agenesis had deciduous molars present; therefore, the presence of agenesis is very hard to be disputed. Peck et al. [14], reported an increased prevalence of premolar agenesis, which was not confirmed by our findings. The presence of any tooth agenesis in orthodontic patients with a suspected maxillary canine impaction may indicate a need for interceptive treatment. Tooth agenesis was not significantly associated with the mandibular canine impaction, except for the mandibular incisor agenesis, but its prevalence was not statistically different between MaxCI and ManCI groups.

### 4.4. Supernumerary Teeth and Tooth Transposition

Supernumerary teeth were significantly more prevalent in both canine impaction groups when compared to the control group and significantly more common in the ManCI group when compared with the MaxCI group. It is reasonable to assume that supernumerary teeth may represent an obstacle for the normal eruption of a tooth, which could be a reason for mandibular canine impaction. The supernumerary teeth in the maxilla are more often located in the incisor region and they are often the cause for maxillary incisor impaction, but it is rarer to see supernumeraries in other parts of a maxilla in nonsyndromic patients.

Tooth transposition is rare with a reported frequency of 0.3% [45]; therefore, the relatively high number of patients with tooth transposition in the ManCI group (12.5%) may be an important predictor for the impaction of the mandibular canine. There were four patients with tooth transposition in the MaxCI group in contrast to one tooth transpositions in the control group.

### 4.5. Transmigration

Transmigration was significantly more common in the ManCI group in comparison with the MaxCI and the control groups and was previously confirmed in the literature as an important predictor for mandibular canine impaction with a prevalence ranging from 0.08% to 0.34% [30,33]. The prevalence of canine transmigration was 18.8% in the ManCI group, which is lower than the prevalence reported by other authors [26,30,31]. This difference can be related to differences in the selected populations and the diagnosis of impaction. Severe canine transmigration can be detected early on panoramic radiographs, as one of the included patients in the ManCI group was 7.5 years old and severe mandibular canine transmigration was already present. However, no preventive treatment measures exist for this condition. It has been documented, that the surgical uprighting (transalveolar transplantation) of severely impacted teeth can be used as a viable treatment option to bring the impacted canine to its normal position [46,47,48], but higher survival and success has been reported for teeth with developing roots [49]. Plakwicz et al. have shown excellent long-term healing and alveolar bone regeneration after a trans-alveolar transplantation of severely impacted developing mandibular canines [50]. The early detection of mandibular canine impaction is therefore very important in the light of a surgical uprighting of the affected canine. Sometimes, orthodontic space opening is needed to accommodate the transplanted canine into a dental arch, which requires additional time before surgery. The detection of associated tooth anomalies in young patients suspected of mandibular canine impaction can be helpful as an important diagnostic tool.

### 4.6. Tooth Impaction

The impaction of other teeth was previously reported in patients with impacted canines, both maxillary [17,20,22] and mandibular canines [27,33]. Tooth impactions were significantly more prevalent in both groups with canine impaction in comparison to the control group and comprised of impactions of second premolars (three), mandibular incisors (three) and maxillary central incisors (two), but no significant differences were found between the MaxCI and ManCI groups. Except for one patient with an impacted mandibular canine and maxillary central incisor, all other patients with impacted canines had impacted teeth located in the same jaw as the impacted canine.

### 4.7. Limitations

Potential sources of bias in studies such as the present one include selection and information bias. We used a sample from individuals seeking orthodontic treatment, and it is possible that the true prevalence in the population may be different than the one recorded in our sample. However, we did include all available patients.

Palatally and buccally impacted canines were combined in one group of patients with maxillary canine impaction (MaxCI). It would be better to separate those two groups with maxillary canine impaction regarding the presence of dental anomalies. However, only 10 canines in our material were buccally impacted, which is too small group for reliable comparisons.

In terms of information bias as it pertains to the accurate assessment of impactions and dental anomalies, for the former impaction, it was confirmed at various stages as outlined in the methods, and the latter was considered a fairly objective outcome to be incorrectly classified.

A larger sample size can help to further evaluate possible associations between a specific anomaly and canine impaction, but the collection of a large sample, especially with mandibular canine impaction is difficult due to the scarcity of cases.

The control group included orthodontic patients without canine impaction from one office, as this was the only dental office from three offices which was limited to orthodontic practice. The aim of the study was to assess dental anomalies in orthodontic patients with canine impaction; therefore, the control patients were not recruited from two other dental offices as a majority of their patients included oral surgery patients.

## 5. Conclusions

Dental anomalies in orthodontic patients with impacted canines are more frequent compared to patients without canine impaction, and their presence may serve as an additional predictive tool in an early detection of canine impaction. Different types of dental anomalies were found in patients with impacted maxillary and mandibular canines, which may indicate that different etiological factors are involved in their occurrence.

## Figures and Tables

**Figure 1 diagnostics-13-02766-f001:**
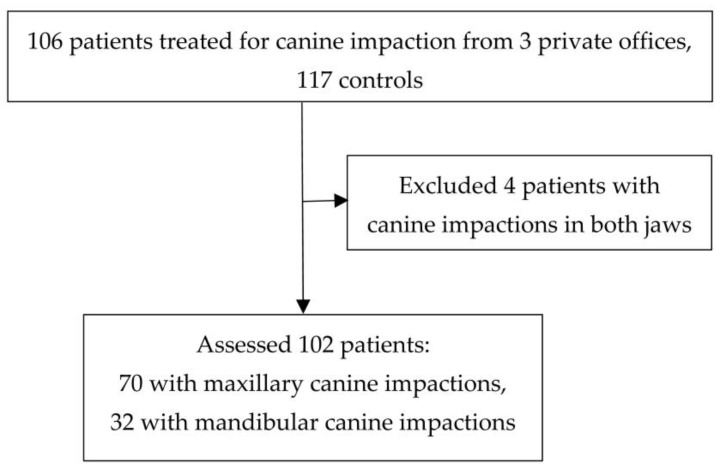
Patients’ flow diagram.

**Figure 2 diagnostics-13-02766-f002:**
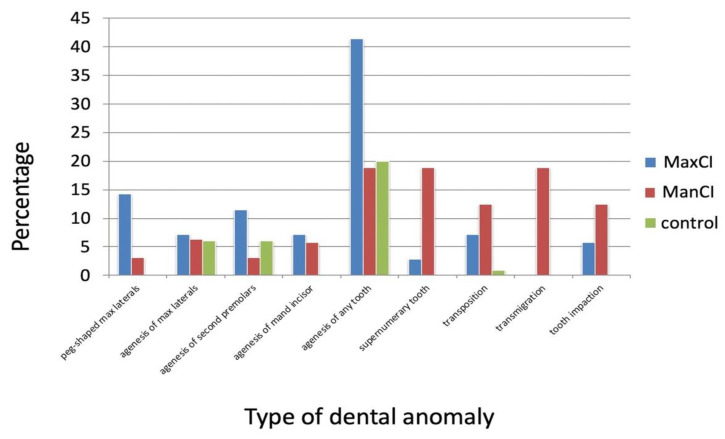
Prevalence of dental anomalies in the MaxCI, the ManCI and in the control groups.

**Table 1 diagnostics-13-02766-t001:** Published studies on maxillary canine impaction and related findings.

Author	Number of Patients/ Number of Impacted Canines	Number of Controls	Dental Anomaly in Total	Peg-Shaped/Small Maxillary Lateral Incisors		Maxillary Lateral Incisor Agenesis		Second Premolars Agenesis		Missing Third Molars		Tooth Agenesis		Transposition		Supernumerary Teeth		Impaction of Other Teeth	
				Impacted Canine	Control Group	Impacted Canine	Control Group	Impacted Canine	Control Group	Impacted Canine	Control Group	Impacted Canine	Control Group	Impacted Canine	Control Group	Impacted Canine	Control Group	Impacted Canine	Control Group
Oliver et al. [11]	60 ^1^		-			3.3% ^1^													
Bjerklin et al. [12]	91 ^1^		-					5.5% ^1^											
Mossey et al. [13]	182 ^1^		-			12.6% ^1^													
Peck et al. [14]	58 ^1^/76 ^1^		-	17.2% ^1^		3.4% ^1^		13.8% ^1^		39.7% ^1^		17.2% ^1^							
Pirinen et al. [15]	106/ 105 ^1^, 1 ^2^			9.4% ^1^								35.8% ^1^		1.9% ^1^		1.9% ^1^			
Chaushu et al. [16]	58 ^1^/58 ^1^	40	18.1% ^4^/ 1.1% ^5^	14.6% ^1^	1.2%	3.5% ^1^	0%												
Leifert et al. [17]	235/281 ^1^	235	-	11.5% ^1^	2.55%	7.2% ^1^	0.85%	11.5% ^1^	8.3%			15.3% ^1^	8.1%					12.8% ^1^	3%
Peck el al. [18]	58 ^1^		57% ^4^			3% ^1^		14% ^1^				40% ^1^							
Jena et al. [19]	66 ^1^		11.8% ^1^/ 38.9% ^2^	16.7% ^1^/ 2.9% ^2^		14.8% ^1^/ 2.9% ^2^													
Mercuri et al. [20]	151/ 114 ^1^, 37 ^2^	151	-	10.5% ^1^/ 8.1% ^2^	2%	6.1% ^1^/ 0% ^2^	2%	2.7% ^1^/ 2.7% ^2^	2.7%					5.3% ^1^/ 0% ^2^	0.7%	0.9% ^1^/ 2.7% ^2^	0%	17.5% ^1^/ 10.8% ^2^	6%
Yan et al. [21]	170/ 69 ^1^, 101 ^2^	170	42.4% ^4^/ 27% ^5^	40.3% ^1^/ 9.9% ^2^	4%	4.3% ^1^/ 0% ^2^	0%			36.2% ^1^/ 30.7% ^2^	23.8%					15.5% ^1^/5.9% ^2^	3%		
Sajnani et al. [22]	533 ^1^		47.5% ^4^	10.1% ^1^/ 6.3% ^2^								11.4% ^1^/ 7.5% ^2^				6.9% ^1^/ 8.2% ^2^		4.6% ^1^/ 10.8% ^2^	
Scerri et al. [23]	477 ^1^	500	78.2% ^4^/ 43.8% ^5^	20.1% ^3^	9% ^3^	20.1% ^3^	9% ^3^	9.2%	6%	37.9%	28.8%			1.9%					
Herrera-Atoche et al. [24]	52 ^1^/65 ^1^	808	52% ^4^/20 ^5^	1.9%	1.4%							7.7%	4.8%	23.1%	1.1%	5.8%	4.9%	19.2%	5.6%

^1^ Palatally displaced canines (PDC); ^2^ Buccally displaced canines (BDC); ^3^ Both peg-shaped and agenesis of maxillary lateral incisor; ^4^ Impacted canine group; ^5^ Control group.

**Table 2 diagnostics-13-02766-t002:** Published studies on mandibular canine impaction and the presence of dental anomalies.

Author	Number of Impacted Mandibular Canines	Material	Prevalence of Canine Impaction	Prevalence of Canine Transmigration	Other Dental Anomalies
Aydin et al. [26]	22	4500 consecutive radiographs	0.44%	0.18%	
Buyukkurt et al. [27]	15	4500 radiographs		0.33%	100% canine transmigration, tooth impaction in 6 patients
Yavuz et al. [28]	65 patients	5022 radiographs	1.29%		
González-Sánchez et al. [29]	15	Retrospective evaluation			100% canine transmigration, 2 supernumerary teeth
Aktan et al. [30]	26	5000 radiographs	0.46%	0.34%	No anomalies except for pathologies such as cysts or odontomas
Celikoglu et al. [31]	9	2215 orthodontic patients	0.4%	0.22%	
Gunduz et al. [32]	86	12,129 radiographs	0.71%		
Kara et al. [33]	85	112,873 radiographs		0.075%	100% canine transmigration; odontomas (4), other impactions (12), transposition (6), second premolar agenesis (4), upper lateral incisor agenesis (3), supernumerary molar (1)
Aras et al. [34]	63 patients, 23 canines evaluated	5100 radiographs	1.35%		
Topkara et al. [35]	14 patients	1527 orthodontic patients	0.92%		
Jain et al. [36]	43 patients	10,422 patients	0.41%		
Kamiloglu and Kelahmet [37]	4	453 orthodontic/ pedodontic patients	0.88%		
Sajnani and King [38]	64	20,347 orthodontic/ pedodontic patients	0.31%		

**Table 3 diagnostics-13-02766-t003:** Number of dental anomalies per patient in patients with MaxCI, ManCI and in controls.

Number of Dental Anomalies per Patient	MaxCI (%)	ManCI (%)	Control (%)
0	33 (47.1)	15 (46.9)	93 (79.5)
1	19 (27.1)	8 (25)	12 (10.3)
2	9 (12.9)	4 (12.5)	8 (6.8)
3	4 (5.7)	3 (9.4)	2 (1.7)
4	1 (1.43)	0 (0)	1 (0.85)
5	1 (1.43)	1 (3.1)	0 (0)
6	1 (1.43)	1 (3.1)	1 (0.85)
8	1 (1.43)		0 (0)
10	1 (1.43)		0 (0)
Total	70 (100)	32 (100)	117 (100)

**Table 4 diagnostics-13-02766-t004:** Comparison of dental anomalies between the groups with impacted maxillary and mandibular canines. In bold are comparisons that reached a statistical significance for MaxCI vs. ManCI (* indicates *p* < 0.05; ** indicates *p* < 0.001).

Anomaly	Location	Prevalence (%)	OR (95% CI)	*p*-Value
Peg-shaped Maxillary Lateral Incisor	Maxillary	10/70 (14.3)	5.11 (0.67, 231.33)	0.17
	Mandible	1/32 (3.1)	Reference	
Agenesis of Maxillary Lateral Incisor	Maxilla	5/70 (7.1)	1.15 (0.18, 12.75)	0.87
	Mandible	2/32 (6.3)	Reference	
Agenesis of Second Premolar	Maxilla	8/70 (11.4)	3.96 (0.49, 183.12)	0.32
	Mandible	1/32 (3.1)	Reference	
Agenesis of Mandibular Incisor	Maxilla	4/70 (7.1)	1.87 (0.18, 95.47)	0.99
	Mandible	1/32 (3.1)	Reference	
Agenesis of Any Tooth	**Maxilla**	29/70 (41.4)	3.03 (1.04, 10.18)	**0.04 ***
	Mandible	6/32 (18.8)	Reference	
Supernumerary Tooth	Maxilla	2/70 (2.9)	0.13 (0.01, 0.79)	**0.02 ***
	**Mandible**	6/32 (18.8)	Reference	
Transposition	Maxilla	4/70 (7.1)	0.43 (0.07, 2.47)	0.42
	Mandible	4/32 (12.5)	Reference	
Transmigration	Maxilla	0/70 (0)	0.05 (0.00, 0.35)	**0.001 ****
	**Mandible**	6/32 (18.8)	Reference	
Tooth impaction	Maxilla	4/70 (5.7)	0.43 (0.07, 2,47)	0.42
	Mandible	4/32 (12.5)	Reference	

**Table 5 diagnostics-13-02766-t005:** Comparison of dental anomalies between the groups with impacted maxillary and mandibular canines and the control group without canine impaction. In bold are comparisons that reached statistical significance for MaxCI vs. control and ManCI vs. control (* indicates *p* < 0.05; **, *** indicates *p* < 0.001).

Anomaly	Location	Prevalence (%)	OR (95% CI)	*p*-Value
Peg-shaped Maxillary Lateral Incisor	**Maxilla**	10/70 (14.3)	26.53 (4.16, +∞)	**<0.001 *****
	Mandible	1/32 (3.1)	3.66 (0.09, +∞)	0.43
	Control	0/117 (0)	Reference	
Agenesis of Maxillary Lateral Incisor	Maxilla	5/70 (7.1)	1.21 (0.29, 4.63)	0.98
	Mandible	2/32 (6.3)	1.05 (0.10, 5.90)	1
	Control	7/117 (6.0)	Reference	
Agenesis of Second Premolar	Maxilla	8/70 (11.4)	1.59 (0.48, 6.88)	0.29
	Mandible	1/32 (3.1)	0.51 (0.01, 4.21)	0.91
	Control	7/117 (6.0)	Reference	
Agenesis of Mandibular Incisor	Maxilla	4/70 (7.1)	9.18 (1.13, +∞)	0.43
	**Mandible**	1/32 (3.1)	3.66 (0.09, +∞)	**0.04 ***
	Control	0/117 (0)	Reference	
Agenesis of Any Tooth	**Maxilla**	29/70 (41.4)	2.87 (1.41, 5.89)	**0.003 ****
	Mandible	6/32 (18.8)	0.94 (0.28, 2.72)	1
	Control	23/117 (20.0)	Reference	
Supernumerary Tooth	Maxilla	2/70 (2.9)	Not estimable as all 0 in MaxCI vs. controls	
	**Mandible**	6/32 (18.8)	34.97 (4.85, +∞)	**<0.001 *****
	Control	0/117 (0)	Reference	
Transposition	Maxilla	4/70 (7.1)	6.96 (0.67, 349.13)	0.13
	**Mandible**	4/32 (12.5)	16.15 (1.52, 822.8)	**0.02 ***
	Control	1/117 (0.9)	Reference	
Transmigration	Maxilla	0/70 (0)	4.08 (0.32, +∞)	0.28
	**Mandible**	6/32 (18.8)	34.97 (4.85, +∞)	**<0.001 ****
	Control	0/117 (0)	Reference	
Tooth impaction	**Maxilla**	4/70 (5.7)	9.18 (1.13, +∞)	**0.04 ***
	**Mandible**	4/32 (12.5)	21.11 (2.56, +∞)	**0.004 ****
	Control	0/117 (0)	Reference	

## Data Availability

The data presented in this study are available upon request from the corresponding author.

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
