# Peer review of "Comparisons of Dental Anomalies in Orthodontic Patients with Impacted Maxillary and Mandibular Canines"

_diagnostics, 2023, doi:10.3390/diagnostics13172766_

Round 1

Reviewer 1 Report

Thank you for submitting this manuscript for review. This is a very important and relevant topic. Your aim is clearly stated.  I have a few comments and suggestions. 

Introduction:

1. Line 36-37 and 64/65 percentages are stated with a comma (,). I would suggest placing a period (.) so it reads as 0.3% or 1.35% for example. 

Methods:

1. In lines 103-105 it states that each panoramic radiograph was assessed for dental anomalies by one person and then confirmed with 2 other authors when necessary.  Was the rater (JS) calibrated and if so how was that done. Who were they calibrated against?  What is the inter and intra rater agreement?  Did JS score a small group for anomalies 2 times separated by time to show consistency and agreement with self?  If not, I would suggest that this be completed. 

What does "when necessary" mean? What are the parameters for having the other two authors rate for anomalies?

2. Why was the control group  only taken from one office? Was there an analysis showing that control patients from the three offices were all the same or similar? How do we know the prevalence of dental anomalies may not be different at the other 2 offices for the controls?  This should be added as a limitation of the study in the discussion. 

Overall nice work on this important topic. It would be great to see this study continued in the future with a larger sample size!

Minor edits

Author Response

Dear Reviewer,

Many thanks for your time to review our paper and for your valuable comments to the manuscript.

  1. We have changed all commas to periods in the Introduction and in the manuscript.
  2. The rater was a young specialist in orthodontics who was calibrated using approximately 70-80 panoramic radiographs of orthodontic patients with dental anomalies with a senior specialist in orthodontics (university teacher and international lecturer) experienced in treatment of patients with tooth agenesis, tooth impaction and dental anomalies with over 25 years of experience. Unfortunately, we have not recorded the exact number of radiographs used for the calibration and the inter and intra rater agreement, but in fact all radiographs included were also assessed by the senior specialist to eliminate any possible errors and discussed with the third rater (PP). We have added in the text some explanation about the rating of radiographs.
  3. The control group included orthodontic patients without canine impaction from one office, as this was the only dental office from three offices, which was limited to orthodontic practice. The aim of the study was to assess dental anomalies in orthodontic patients with canine impaction, therefore the control patients were not recruited from two other dental offices as majority of their patients included oral surgery patients. The orthodontic practice from which was recruited a control group has over 5000 orthodontic patients and was estimated to be representative. This information was added in the limitations.

Reviewer 2 Report

The authors perform an analysis regarding the presence and possible associations between type of dental anomalies and maxillary and mandibular canine impaction.

The manuscript is well written and organized.

The design of the study is adequately described in the materials and methods section.

 The results are clearly presented and the conclusions of the paper are supported by them.

The authors should improve the quality of Figure 1.

Page 12 line 356: this line should be deleted.

Author Response

Dear Reviewer,

Many thanks for your time to review our paper and for your valuable comments to the manuscript.

We have improved the quality of the Figure 1 and deleted the line 356 as requested and it was our mistake. 

Round 2

Reviewer 1 Report

Thank you for your responses. My questions and comments have been addressed.